# Fusion of histone variants to Cas9 suppresses non-homologous end joining

**Tomoko Kato-Inui[1], Gou Takahashi[ID][1], Terumi Ono[1,2], Yuichiro Miyaoka[ID][1,2,3]***

**1** Tokyo Metropolitan Institute of Medical Science, Regenerative Medicine Project, Tokyo, Japan,
**2** Graduate School of Medical and Dental Sciences, Tokyo Medical and Dental University, Tokyo, Japan,
**3** Graduate School of Humanities and Sciences, Ochanomizu University, Tokyo, Japan

* miyaoka-yi@igakuken.or.jp

**Data Availability Statement:** All relevant data are within the paper and its Supporting information files.

**Funding:** This work was supported by Japan Society for the Promotion of Science KAKENHI

## Abstract

As a versatile genome editing tool, the CRISPR-Cas9 system induces DNA double-strand breaks at targeted sites to activate mainly two DNA repair pathways: HDR which allows precise editing via recombination with a homologous template DNA, and NHEJ which connects two ends of the broken DNA, which is often accompanied by random insertions and deletions. Therefore, how to enhance HDR while suppressing NHEJ is a key to successful applications that require precise genome editing. Histones are small proteins with a lot of basic amino acids that generate electrostatic affinity to DNA. Since H2A.X is involved in DNA repair processes, we fused H2A.X to Cas9 and found that this fusion protein could improve the HDR/NHEJ ratio by suppressing NHEJ. As various post-translational modifications of H2A.X play roles in the regulation of DNA repair, we also fused H2A.X mimicry variants to replicate these post-translational modifications including phosphorylation, methylation, and acetylation. However, none of them were effective to improve the HDR/NHEJ ratio. We further fused other histone variants to Cas9 and found that H2A.1 suppressed NHEJ better than H2A.X. Thus, the fusion of histone variants to Cas9 is a promising option to enhance precise genome editing.

## Introduction

Histones are proteins that constitute eukaryotic chromosomes and have five subtypes: H1, H2A, H2B, H3, and H4. The four subtypes except H1 constitute the core histones, and two molecules of each (H2A-H2B and H3-H4) are assembled to form the histone octamer [1]. Histones are characterized by a high content of positively charged amino acids (lysine and arginine) to bind to DNA molecules. DNA wraps around the surface of each histone octamer, which constitutes the nucleosome, the smallest unit of chromatin structure. This is the initial step of DNA folding when DNA is packed into the nucleus. Furthermore, histones undergo various post-translational modifications. In particular, the serine, lysine, and arginine residues of histone tails, the N-terminal site of nucleosomal histones, are known to be subject to phosphorylation, acetylation, methylation, and ubiquitination [2].

H2A, H2B, and H3 have variants that differ in amino acid sequence by a few to several tens of percent from the canonical histones. Many of these histone variants remain

(Grant Number 19K06631), Takeda Science Foundation, Uehara Memorial Foundation (to T.K-I.); Japan Society for the Promotion of Science KAKENHI (Grant Number 17H04993 and 20H03442) (to Y.M). The funders had no role in study design, data collection and analysis, decision to publish, or preparation of the manuscript.

**Competing interests:** The authors have declared that no competing interests exist.

uncharacterized, but some variants alter chromatin dynamics through their incorporation into specific chromatin regions [3] and are involved in various biological processes such as DNA repair, heterochromatin formation, DNA replication, and transcriptional regulation [4]. H2A. X, one of the H2A variants, is phosphorylated at the serine (S) 139 by the ataxia-telangiectasia mutated kinase (ATM), allowing the formation of γH2A.X (H2A.X phosphorylated at S139) in response to DNA double-strand breaks (DSBs) [5]. Then, mediator of DNA damage protein checkpoint protein 1 (MDC1) binds to γH2A.X to initiate the DNA repair process by recruiting various DNA repair factors [6]. K134 dimethylation by the histone methyltransferase SUV39H2 is also correlated with γH2A.X. The K134A mutation that prevents this dimethylation reduces the expression of γH2A.X [7]. In addition, H2A.X acetylated at the lysine (K) 5 by TIP60 histone acetylase is released from chromatin in DNA damage sites and binds to DNA damage response factors to modulate DNA repair response [8–10]. Thus, various post-translational modifications of histones play important roles in DNA repair.

DNA repair in response to DSBs mainly relies on two pathways: homology-directed repair (HDR) mediated by recombination with a homologous template that yields precise repair products identical to the DNA sequence of the template, and non-homologous end joining (NHEJ) that brings the two broken DNA ends together often with random insertions or deletions [11]. However, mammalian cells preferentially adopt NHEJ over HDR by the following mechanisms: NHEJ is active through the cell cycle, whereas HDR is restricted to the S/G2 phases; NHEJ is faster than HDR [12]. We have observed the same trend in genome editing by CRISPR-Cas9 [13]. Therefore, strategies to enhance HDR over NHEJ are required.

Here, we fused H2A.X to Cas9 to see if the HDR and NHEJ activities could be altered. In addition, since the post-translational modifications of H2A.X have been implicated in DNA repair, we examined whether mimicry mutations of these modifications could further improve the HDR/NHEJ ratio. We also fused other H2A and H3 variants to Cas9 and found that some of them improved the HDR/NHEJ ratio by suppressing NHEJ.

## Materials and methods

### Statistical analysis

The transfection experiments were performed in triplicates (three biological replicates). Statistical significance was assessed by a two-tailed Student's *t*-test to compare the differences between two different conditions.

### Plasmids and single-stranded DNA (ssDNA) donor

pCP (expression plasmid of the puromycin-resistant gene and Cas9, Addgene plasmid #204743) was derived from PX459 V2.0 (Addgene plasmid #62988) by removing the human U6 promoter and the gRNA scaffold from it. For N-terminal fusion of H2A.X to Cas9, a GGGGS linker was inserted between the open reading frame of H2A.X and Cas9 sequence of pCP (H2A.X-GS-Cas9 (N-GS, S1 Fig)). Then, H2A.X-GS3-Cas9 (N-GS3) and H2A. X-GS5-Cas9 (N-GS5) were generated by inserting additional linkers between the original GGGGS linker and Cas9 sequence of N-GS, respectively. For the C-terminal fusion of H2A.X to Cas9, the open reading frame of H2A.X and GGGGS linkers were inserted at the C-terminus of Cas9 to generate Cas9-GS-H2A.X (C-GS, S1 Fig), Cas9-GS3-H2A.X (C-GS3), and Cas9-GS5-H2A.X (C-GS5), respectively. The mimicry and inhibitory mutations for acetylation, phosphorylation, and methylation of H2A.X were introduced by inverse PCR into N-GS3. The cDNAs of H2A, H2A variants, H2B, H3, and H3 variants were cloned into the N-GS3 backbone plasmid. pGB (expression vector of the blasticidin-resistant gene and guide RNA (gRNA), Addgene plasmid #204744) was derived from PX459 V2.0 by exchanging the

Cas9 open reading frame for that of the blasticidin-resistant gene (S1 Fig). Oligonucleotides with the gRNA sequence were cloned into pGB in the same way as for PX459 V2.0. ssDNA donors used in this study were ultramer DNA oligonucleotides (FASMAC, Kanagawa, Japan). The sequences of the primers used in this study are shown in S1 Table.

## HEK293FT cell culture, transfection, and genomic DNA extraction

Human embryonic kidney (HEK) 293FT cell line was maintained in DMEM medium (Nacalai Tesque, Kyoto, Japan) supplemented with 10% fetal bovine serum (JRH Biosciences, Lenexa, KS, USA) and 100 µg/ml penicillin-streptomycin (Thermofisher Scientific, Waltham, MA, USA) at 37˚C with 5% $CO_2$. HEK293FT cells were plated at 30,000 cells/well in a 96-well plate one day before transfection. Transfection was performed with Lipofectamine 2000 (Thermo-fisher Scientific) according to the manufacturers' instructions. Forty-five ng/well of an expression plasmid for Cas9 (pCP) or Cas9 tethered with histone variants, 45 ng/well of a gRNA expression plasmid (pGB), and 10 ng/well of ssDNA donor were transfected. The next day, 5 µg/ml of puromycin and 100 µg/ml of blasticidin were added to select cells transfected with both the Cas9-expression and gRNA-expression plasmids. Three days after transfection, genomic DNA was extracted as previously described [14]. Briefly, survived cells were resuspended in 50 µl/well of genomic lysis buffer (0.01M Tris-Cl at pH7.5, 0.02M EDTA at pH8.0, 0.01M NaCl, 0.5% N-Lauroylsarcosine sodium salt and 0.1 mg/ml Proteinase K) at 55˚C overnight, and the genome DNA was precipitated by using 100% ethanol with 0.075M NaCl buffer. The precipitated DNA was rinsed with 70% ethanol and then dried up. The genomic DNA was resuspended in 30 µl/well of water. Target genes and mutations engineered in this study are shown in S2 Table. The sequences of ssDNA donors used in this study are shown in S3 Table.

## Digital PCR assay to detect the HDR and NHEJ activities

The digital PCR-based assay to detect the HDR and NHEJ activities was described previously [13]. To prepare samples for digital PCR, 100–180 ng of genomic DNA, 12 µl of 2×ddPCR Supermix for Probes (no dUTP) (Bio-Rad Laboratories, Hercules, CA, USA), 0.6 µl of primers and probe sets (S5 Table), 0.48 µl of restriction enzyme for fragmentation (HindIII for RBM20, ATP7B, and APOE; RspRSII for GRN) were mixed with water added up to 24 µl per sample. Nano-litter scale droplets containing the PCR reagents were generated using the Droplet Generator (Bio-Rad Laboratories) and transferred to an Eppendorf twin.tec PCR plate (Eppendorf, Hamburg, Germany). The plate was covered with a PCR Plate Heat Seal, foil, pierceable (Bio-Rad Laboratories) using a PX1 PCR Plate Sealer (Bio-Rad Laboratories) according to the manufacturer's instructions. Thermal cycling was performed using the C1000 Touch Thermal Cycler (Bio-Rad Laboratories) and then the fluorescent signals of the droplets were analyzed using the QX200 Droplet Reader (Bio-Rad Laboratories). The assay was designed to detect the wild-type, the HDR, and the NHEJ alleles as FAM and HEX double-positive, FAM highly-positive, and FAM single-positive populations, respectively (S2 Fig). The frequency of the HDR and NHEJ alleles was calculated and converted to a superimposed bar graphs and dot plots to represent the data. The sequences of RBM20-2, RBM20-g1, GRN-2, and GRN-g2, ATP7B-3, ATP7B-g3, and APOE gRNAs used in this study are shown in S4 Table. Assay components used for digital PCR are shown in S5 Table. Thermal cycle conditions for digital PCR are shown in S6 Table.

## Analysis of off-target effects

Three potential off-target sites for each gRNA were identified by CRISPOR (http://crispor.org) [15] based on the protospacer adjacent motif (PAM) sequence in the hg38 reference genome

and a high mitOfftargetScore (S7 Table). Off-target editing frequencies in HEK293FT cells were measured by targeted amplicon sequencing. The same genomic DNA samples used in the digital PCR analysis were used for amplicon sequencing. Amplification of off-target genomic regions and addition of adapters and indexes were conducted through two-step PCR. The library preparation was performed with the same protocol as the NGS analysis previously performed in our laboratory [16]. All libraries were mixed in 4 nM amounts and 20% PhiX Control v3 (illumina) was added for amplicon sequencing. Sequencing was performed with MiSeq (illumina) using MiSeq Reagent Micro Kit v2 (illumina) according to the manufacturer's instructions. Primers and PCR condition used for off-target detection were shown in S8–S10 Tables. Fastq files generated by MiSeq were imported into the CLC Genomics Workbench (QIAGEN, Hilden, Germany) and adapter sequences were trimmed and demultiplexed using the index sequences. The data were analyzed by CRISPResso2 (https://github.com/pinellolab/CRISPResso2) [17] in the CRISPResso Batch mode, described previously [16]. The assay background was negligible at less than 0.1% for all gRNAs (S11 and S15 Tables).

## Results

### Improvement of the HDR/NHEJ ratio by fusion of H2A.X to Cas9

As H2A.X initiates DNA repair, we hypothesized that the HDR/NHEJ ratio induced by Cas9 could be altered by bringing H2A.X close to the cleavage site by protein fusion. Therefore, we N-terminally tethered H2A.X to Cas9 via the GGGGS linker (N-GS) (Fig 1A). By using N-GS and guide RNAs (gRNAs) previously designed (RBM20-2, RBM20-g1, GRN-2, and GRN-g2, [14]), we introduced two pathogenic point mutations: RBM20 R636S and GRN R493X in HEK293FT cells (Fig 1B). We found that N-GS induced less NHEJ compared to the normal Cas9 while keeping the HDR level comparable with RBM20-2 and GRN-g2, resulting in the increased HDR/NHEJ ratio by 1.4-fold compared to the normal Cas9 (Fig 1C, S11 Table). These results indicated that the fusion of H2A.X to Cas9 could enhance the HDR/NHEJ ratio and prompted us to further optimize the design of the fusion.

### Optimization of the length and position of the linker to tether H2A.X to Cas9

Since a flexible linker can have a profound effect on fusion protein stability and activity [18,19], we optimized the length and position of the fusion. We tested GGGGS, (GGGGS)$_3$, or (GGGGS)$_5$ linker to tether H2A.X to the N- or C-terminus of Cas9 (Fig 1A). We named these fusion proteins N-GS, N-GS3, N-GS5, C-GS, C-GS3, and C-GS5 depending on the length and position of the linkers (Fig 1A). We examined the HDR and NHEJ activities of these fusion proteins. We found that most of the fusion proteins suppressed NHEJ compared to using Cas9 alone, but N-GS3 improved the HDR/NHEJ ratio the most (1.2- to 1.5-fold increase) (Fig 1C, S11 Table). Therefore, we decided to further modify and improve N-GS3.

### Mimicry variants of H2A.X S139 phosphorylation or K134 methylation did not improve the HDR/NHEJ ratio

The post-translational modifications of H2A.X have been reported to be involved in DNA damage repair [5,20]. In particular, γH2A.X (H2A.X phosphorylated at S139) is the most well-known marker of DNA damage and functions as a platform for the recruitment of DNA damage response (DDR) signaling factors, but its specific involvement in the HDR and/or NHEJ pathways has not yet been reported. Therefore, we generated an S139D phosphorylation mimic mutant of H2A.X fused to Cas9 (SD-Cas9, Fig 2A). However, the HDR and NHEJ

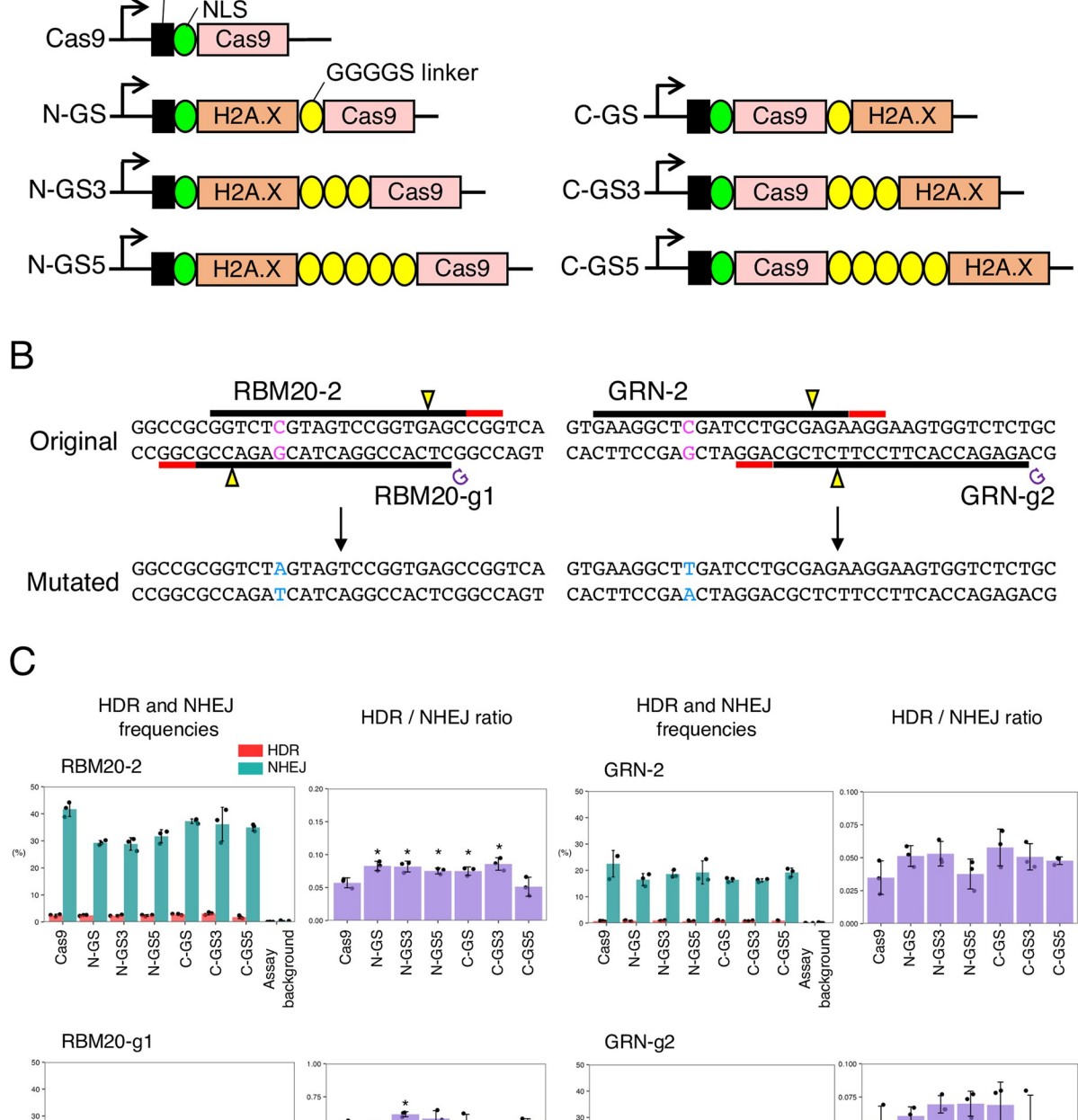

**Fig 1. HDR and NHEJ activities of fusion proteins of H2A.X and Cas9 in HEK293FT cells.** A. A schematic representation of N-terminal and C-terminal fusion of H2A.X and Cas9. The arrows indicate the transcription start sites. B. The genomic sequences around the targeted point mutations and designed gRNAs in RBM20 and GRN. Protospacer adjacent motifs (PAMs), cleavage sites, and targeted and substituted nucleotides are represented by red lines, yellow triangles, and magenta and light blue characters, respectively. C. The HDR and NHEJ activities of fusion proteins of H2A.X and Cas9 in HEK293FT cells with RBM20-2, RBM20-g1, GRN-2, and GRN-g2 gRNAs. Means ± S.E. of the frequencies of HDR alleles (red) and NHEJ alleles (green) are shown (n = 3) on the left, and means ± S.E. of the HDR/NHEJ ratio are shown (n = 3) on the right. Student's *t*-test was used to evaluate the difference in the HDR and NHEJ activities between Cas9 alone and the fusion proteins. *$P<0.05$ and **$P<0.01$.

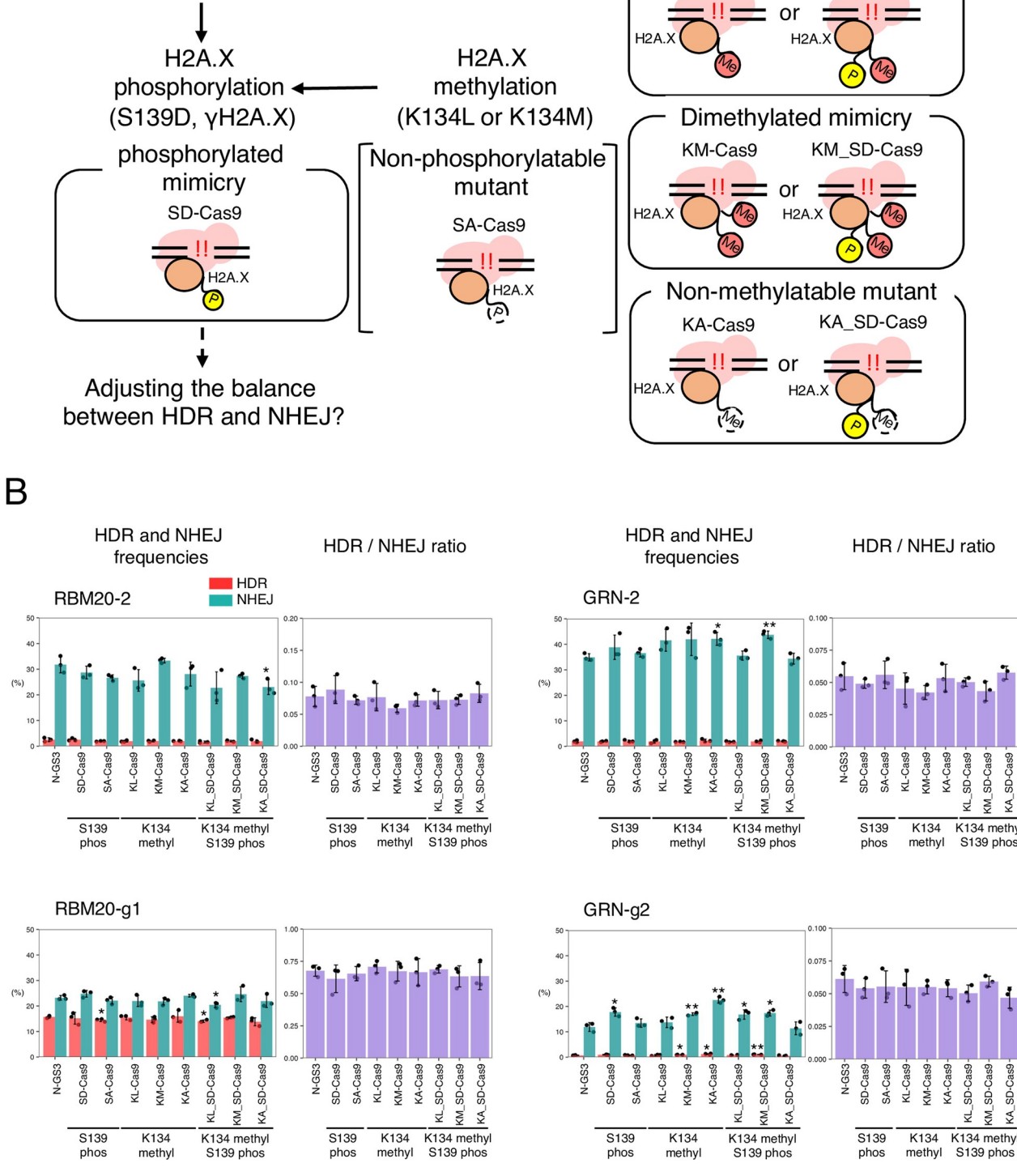

**Fig 2. HDR and NHEJ activities of fusion proteins of Cas9 and H2A.X with mimicry mutations of phosphorylation and methylation.** A. A schematic representation of the cellular response to double-strand breaks involving phosphorylated and methylated H2A.X. Yellow "P", and orange "Me", with solid lines indicate the mimicry mutations for phosphorylation and methylation, respectively. White "P" and "Me" with dashed lines indicate the non-phosphorylatable and non-methylatable mutations, respectively. B. The HDR and NHEJ activities of the fusion proteins of Cas9 and H2A.X with mutations related to phosphorylation and methylation. Means ± S.E. of the frequencies of HDR alleles (red) and NHEJ alleles (green) are shown (n = 3) on the left, and means ± S.E. of the HDR/NHEJ ratio are shown (n = 3) on the right. Student's *t*-test was used to evaluate the difference in the HDR and NHEJ activities between N-GS3 and the fusion proteins. *$P<0.05$ and **$P<0.01$.

activities of SD-Cas9 were comparable to those of N-GS3 with RBM20-2, RBM20-g1, and GRN-2. With GRN-g2, both HDR and NHEJ activities of SD-Cas9 were increased compared to N-GS3, but the HDR/NHEJ ratio was still comparable (Fig 2B, S12 Table). We also fused Cas9 and H2A.X with the S139A non-phosphorylatable mutation (SA-Cas9), but the HDR/ NHEJ ratio was not significantly altered by SA-Cas9 either (Fig 2B, S12 Table). It has been reported that dimethylated K134 is critical for H2A.X S139 phosphorylation [7], although it is still debatable [21]. Therefore, we addressed whether mimicries of the H2A.X methylation at K134 improve the HDR/NHEJ ratio. We mutated K134 of H2A.X to leucine (K134L, KL) as a monomethylated mimicry, to methionine (K134M, KM) as a dimethylated mimicry, and to alanine (K134A, KA) as a non-methylatable mutant, respectively. In addition, to validate the synergistic function of methylation at K134 and phosphorylation at S139, we combined the KL, KM, and KA mutants to SD-Cas9 to generate KL_SD-Cas9, KM_SD-Cas9, and KA_SD--Cas9, respectively (Fig 2A). We measured the HDR and NHEJ activities of these fusion proteins with the four gRNAs. However, the HDR/NHEJ ratios of all the fusion Cas9s with H2A.X with the post-translational modification mimic mutations were comparable (0.7-fold decrease to 1.1-fold increase) to that of N-GS3 (Fig 2B, S12 Table).

## Mimicry variants of H2A.X K5 acetylation did not improve the HDR/NHEJ ratio

H2A.X acetylated at K5 recruits DNA repair proteins to the DNA damage sites by binding to DDR signaling factors [8]. Moreover, it has been shown that inhibition of acetylation prevents the accumulation of the DNA repair factors [8]. To investigate whether H2A.X mimicries of acetylation of K5 can alter the balance of HDR and NHEJ, we mutated K5 to glutamine (K5Q, KQ) as an acetylation mimicry variant, and K5 to arginine (K5R, KR) as a non-acetylatable H2A.X variant, respectively (Fig 3A). We found that the NHEJ activity in KQ-Cas9 was slightly increased compared to N-GS3 with GRN-2 and GRN-g2 gRNAs, but no such trend was observed with RBM20-2 and RBM20-g1 gRNAs. The NHEJ activity in KR-Cas9 was slightly decreased compared to N-GS3 with RBM20-g1, but this trend was not observed with the other gRNAs (Fig 3B, S13 Table). These results overall indicate that the fusion of Cas9 with a mimicry of K5 acetylation or a non-acetylatable variant of H2A.X did not result in an improvement of the HDR/NHEJ ratio (0.8-fold decrease to 1.1-fold increase compared to N-GS3, S13 Table).

## Fusion of H2A variants and Cas9 suppressed NHEJ

It is known that histone H2A has several variants. Among the H2A variants, H2A.1, H2A.2, H2A.L, and H2A.J differ from H2A by only a few amino acid residues, whereas H2A.Z, macroH2A.1, and H2A.B have less than 50% amino acid homology to H2A. H2A.X is considerably different from H2A at their C-terminal sequences but is otherwise similar to H2A (Fig 4A). These histone variants have been reported to regulate chromatin structure and gene expression by replacing canonical histones [22]. To examine whether H2A variants and H2B improve the HDR/NHEJ ratio, we fused each of those molecules to Cas9 in the same manner as shown in Fig 1A, N-GS3. Among these Cas9 fusions with the H2A variants and H2B, H2A.1-Cas9 showed decreased NHEJ with RBM20-2, RBM20-g1, and GRN-2 gRNAs, but comparable HDR with RBM20-2 and GRN-2 gRNAs compared to N-GS3, although the increase in the HDR/NHEJ ratio was not statistically significant (Fig 4B, S14 Table).

We also tested Cas9 fusion proteins with the H3 variants. As for the H3 variants, H3.1, H3.2, and H3.3 differ from H3 by only a few amino acid residues (Fig 4C). Compared to

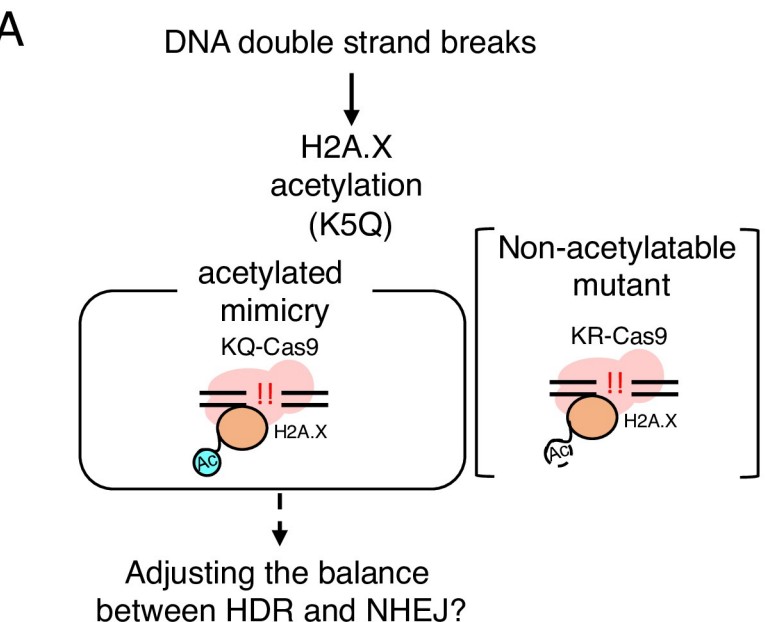

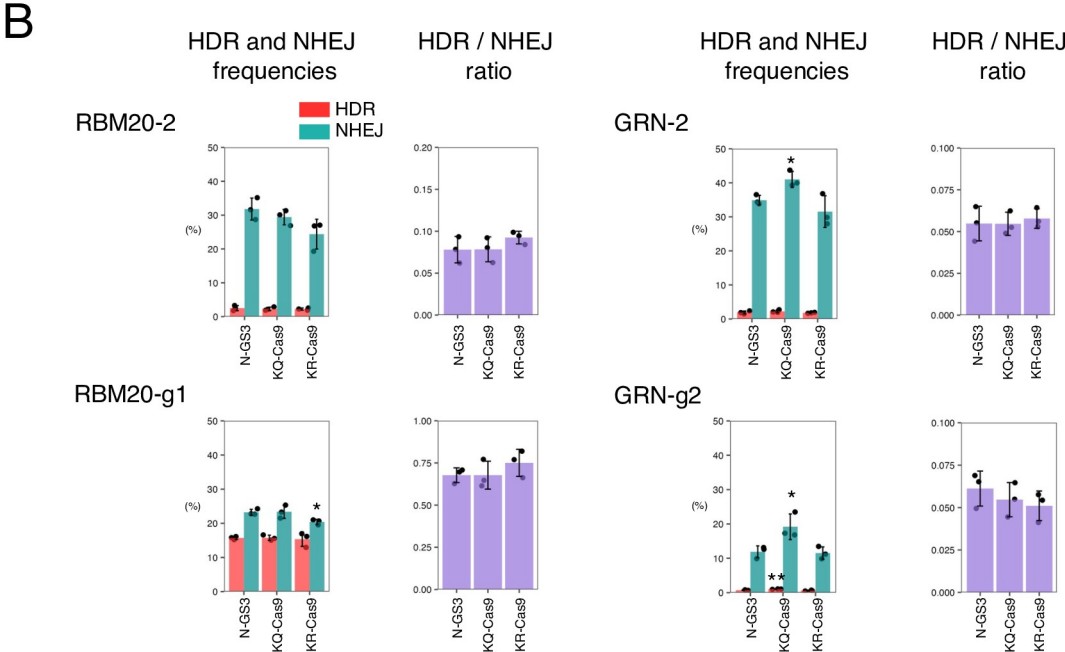

**Fig 3. HDR and NHEJ activities of fusion proteins of Cas9 and H2A.X with acetylation mimicry or inhibitory mutations.**
A. A schematic representation of the cellular response to double-strand breaks involving the acetylated H2A.X. Light blue "Ac" with a solid line and white "Ac" with a dashed line indicate the acetylation mimicry mutation and the non-acetylatable mutation, respectively. B. The HDR and NHEJ activities of the fusion proteins of Cas9 and H2A.X with the acetylation-related mutations. Means ± S.E. of the frequencies of HDR alleles (red) and NHEJ alleles (green) are shown (n = 3) on the left, and means ± S.E. of the HDR/NHEJ ratio are shown (n = 3) on the right. Student's *t*-test was used to evaluate the difference in the HDR and NHEJ activities between N-GS3 and the fusion proteins. *$P<0.05$ and **$P<0.01$.

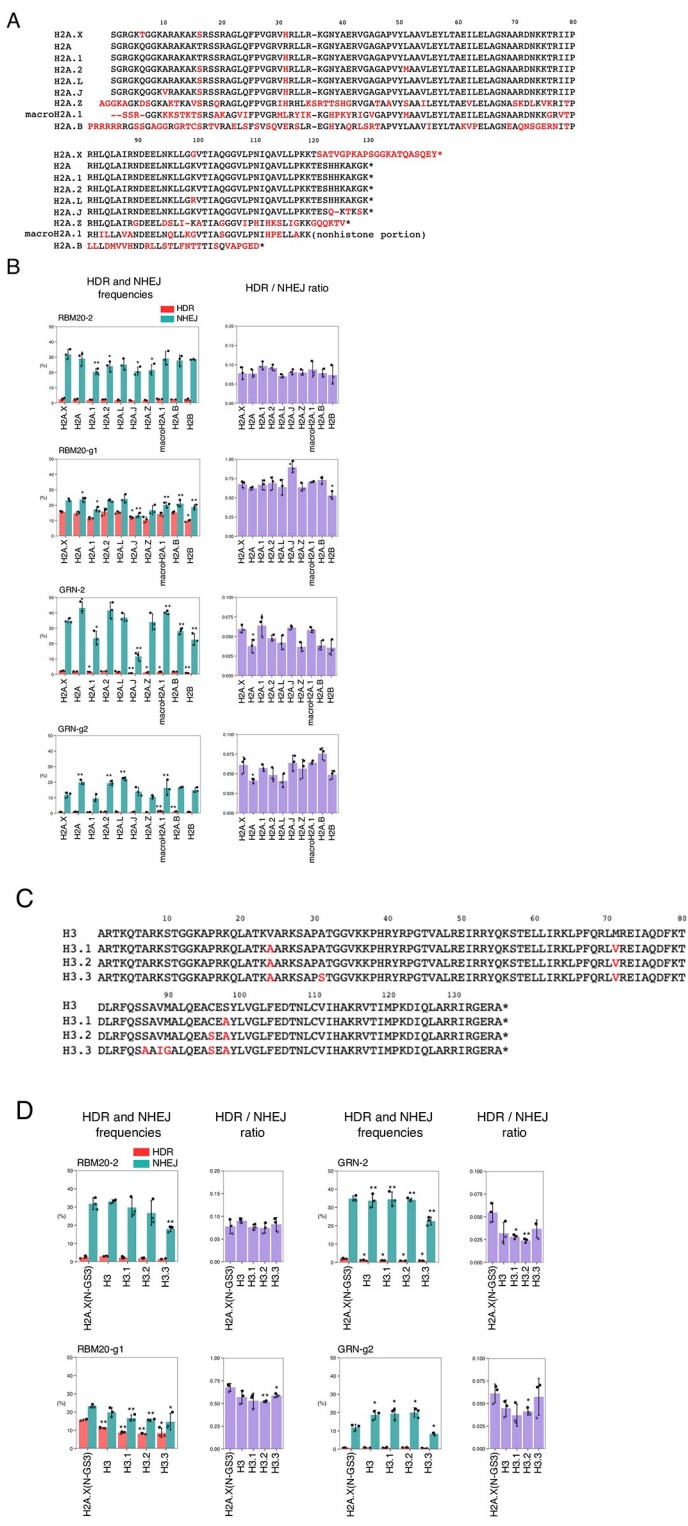

**Fig 4. HDR and NHEJ activities of fusion proteins of H2A, H2B, and H3 variants and Cas9.** A. Amino acid sequences of H2A and H2A variants. Red letters indicate amino acid sequence differences from H2A. Hyphens (-) and asterisks (*) represent missing amino acids and terminating codons, respectively. B. The HDR and NHEJ activities of fusion proteins of Cas9 and H2A variants or H2B. Means ± S.E. of the frequencies of HDR alleles (red) and NHEJ alleles (green) are shown (n = 3) on the left, and means ± S.E. of the HDR/NHEJ ratio are shown (n = 3) on the right. Student's *t*-test was used to evaluate the difference in the HDR and NHEJ activities between N-GS3 and the other

fusion proteins. *$P<0.05$ and **$P<0.01$. C. Amino acid sequences of H3 and H3 variants. Red letters indicate amino acid sequence differences from H3. Hyphens (-) and asterisks (*) represent missing amino acids and terminating codons, respectively. D. The HDR and NHEJ activities of fusion proteins of Cas9 and H3 variants. Means ± S.E. of the frequencies of HDR alleles (red) and NHEJ alleles (green) are shown (n = 3) on the left, and means ± S.E. of the HDR/NHEJ ratio are shown (n = 3) on the right. Student's *t*-test was used to evaluate the difference in HDR and NHEJ activities between N-GS3 and the other fusion proteins. *$P<0.05$ and **$P<0.01$.

N-GS3, H3.3 showed decreased NHEJ with all gRNAs but HDR was also decreased (Fig 4D, S14 Table).

## Fusion of H2A.X or H2A.1 to Cas9 suppresses NHEJ with three other gRNAs

To address whether the fusion of the H2A variants and Cas9 can be a general strategy to suppress NHEJ, we measured the activities of N-GS3 (H2A.X-Cas9) and H2A.1-Cas9 with ATP7B-3 and ATP7B-g3 gRNAs in ATP7B, and APOE-g1 gRNA in APOE (Fig 5A). We observed that both N-GS3 and H2A.1-Cas9 suppressed NHEJ while keeping a comparable level of HDR with three gRNAs compared to Cas9 alone (Fig 5B, S15 Table). Based on these results (Figs 4 and 5), we concluded that H2A.1 is the best option of the histone variants tested in this study to fuse with Cas9 for achieving the highest HDR/NHEJ ratio mainly by suppressing NHEJ. Therefore, we quantified the frequencies of off-target effects at predicted potential off-target sites induced by Cas9 and H2A.1-Cas9. We found that the frequencies of off-target events were comparable between H2A.1-Cas9 and Cas9 (S3 Fig, S16 Table). These results indicate that the fusion of histone variants to Cas9 does not affect its specificity.

## Discussion

In this study, we initially found that H2A.X tethered to Cas9 with GGGGS linkers improved the HDR/NHEJ ratio compared to Cas9 alone. Therefore, we investigated whether mimicries of post-translational modifications of H2A.X could further improve the HDR/NHEJ ratio, but none of them were effective. However, we found that the H2A.1 variant suppressed NHEJ better than H2A.X when fused to Cas9. There have been several reports of the fusion of HDR factors with Cas9 to increase the HDR activity [19,23–27], but this is the first report that the fusion of histones to Cas9 can improve the HDR/NHEJ ratio mainly by suppressing NHEJ.

S139 phosphorylated H2A.X (γH2A.X) rapidly accumulates at the sites of DNA damage and plays a role in DNA repair [5,20]. Therefore, in this study, we generated a mimicry variant of γH2A.X by substituting S139 with an aspartic acid and fusing it to Cas9 (SD-Cas9) (Fig 2A). However, unfortunately, we found that SD-Cas9 did not improve the HDR/NHEJ ratio (Fig 2B, S12 Table). We also examined whether the fusion of Cas9 and H2A.X methylation mimicry improves the HDR/NHEJ ratio since it is known that K134 of H2A.X is dimethylated by SUV39H2, resulting in the γH2A.X production [7]. Therefore, we generated fusion proteins of Cas9 with H2A.X variants of K134 methylation mimicry and non-methylatable mutant of K134 (Fig 2A). However, these fusion proteins were not effective compared to N-GS3 either (Fig 2B, S12 Table).

In addition, since acetylation at K5 of H2A.X is important for assembling DNA repair proteins to damaged sites [8], we tested whether the fusion of Cas9 with mimicry of K5 acetylation of H2A.X, playing this role, could enhance the HDR activity. Contrary to our expectations, however, the fusion of Cas9 with mimicry of K5 acetylation did not directly improve the HDR/NHEJ ratio. Acetylation at K5 and phosphorylation at S139 of H2A.X are important components of the cellular response to DNA damage. Further studies are necessary to

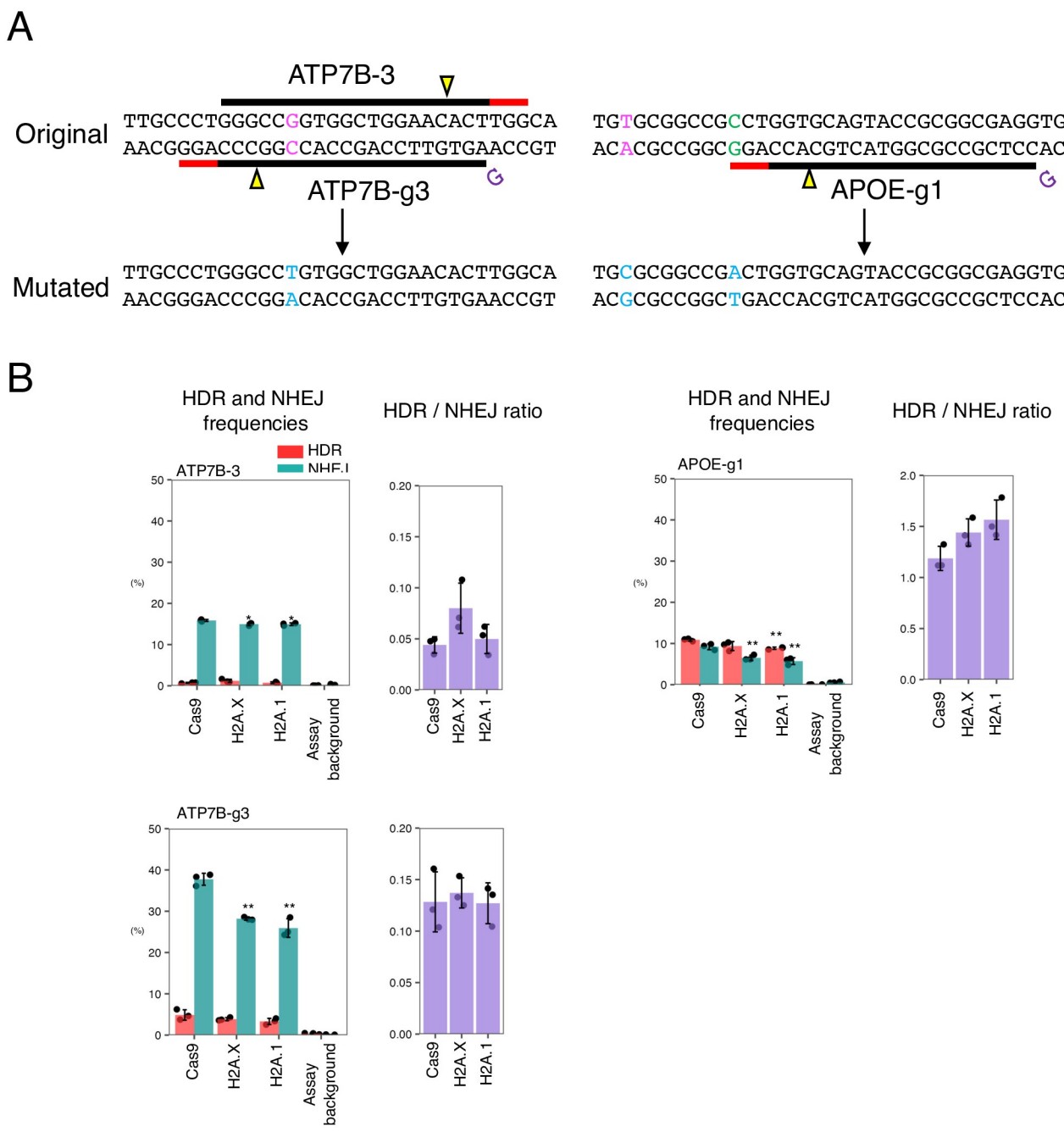

**Fig 5. HDR and NHEJ activities of N-GS3 and H2A.1-Cas9 in ATP7B and APOE.** A. The genomic sequences around the targeted point mutations and designed gRNAs in ATP7B and APOE. PAMs, cleavage sites, as well as targeted nucleotides, synonymous and nonsynonymous nucleotides are represented by red lines, yellow triangles, and characters in magenta, green, and light blue, respectively. B. The HDR and NHEJ activities of Cas9, fusion proteins of Cas9 and H2A.X or Cas9 and H2A.1. Means ± S.E. of the frequencies of HDR alleles (red) and NHEJ alleles (green) are shown (n = 3) on the left, and means ± S.E. of the HDR/NHEJ ratio are shown (n = 3) on the right. Student's *t*-test was used to evaluate the difference in the HDR and NHEJ activities between Cas9 and the other fusion proteins. *P<0.05 and **P<0.01.

understand how these post-translational modifications of H2A.X are involved in DNA repair and apply this knowledge to improving precise genome editing.

As mentioned above, H2A.X was known to accumulate at damaged DNA sites after phosphorylation and be responsible for DNA repair, but little is known about other histone variants. In this study, we discovered that H2A.1 induced less NHEJ while keeping a comparable level of HDR to other histone variants when fused to Cas9 (Figs 4 and 5, S14 and S15 Tables). We also found that the specificity of genome editing was not influenced by fusion of Cas9 to H2A.1 (S3 Fig, S16 Table). The component ratios of H2A.1 and H2A.2 are known to change with aging and differentiation in rat liver tissue and human fibroblasts [28,29]. However, the improvement of the HDR activity with H2A.1 was found for the first time in this study, suggesting a previously unknown role in DNA repair for this histone variant.

In conclusion, we found that the fusion of histone variants H2A.1 to Cas9 suppresses its NHEJ activity. These findings will lead to the development of more precise genome editing platforms.

## Supporting information

**S1 Fig. The sequences of H2A.X-GS-Cas9 (N-GS), Cas9-GS-G2A.X(C-GS), and pGB.** (DOCX)

**S2 Fig. Design of the assay to simultaneously detect HDR and NHEJ at the *APOE* locus.** (PDF)

**S3 Fig. Frequencies of off-target events for each gRNA used in this study.** (PDF)

**S1 Table. Oligonucleotides used for plasmid constructions in this study.** (PDF)

**S2 Table. Target genes and mutations engineered in this study.** (PDF)

**S3 Table. Oligonucleotide donor DNAs used in this study.** (PDF)

**S4 Table. gRNAs used in this study.** (PDF)

**S5 Table. Assay components used for digital PCR in this study.** (PDF)

**S6 Table. Thermal cycle conditions of digital PCR.** (PDF)

**S7 Table. Off-target sites identified by using the CRISPOR web tool.** (PDF)

**S8 Table. Oligonucleotides for the first PCR of amplicon sequencing.** (PDF)

**S9 Table. Oligonucleotides for the second PCR of amplicon sequencing.** (PDF)

**S10 Table. Reagent composition and thermal cycle conditions of PCR for preparation of libraries for amplicon sequencing.**
(PDF)

**S11 Table. Digital PCR raw data of Fig 1C.**
(PDF)

**S12 Table. Digital PCR raw data of Fig 2B.**
(PDF)

**S13 Table. Digital PCR raw data of Fig 3B.**
(PDF)

**S14 Table. Digital PCR raw data of Fig 4B and 4D.**
(PDF)

**S15 Table. Digital PCR raw data of Fig 5B.**
(PDF)

**S16 Table. Frequency of modified sequences detected at off-target loci.**
(PDF)

## Author Contributions

**Conceptualization:** Tomoko Kato-Inui, Yuichiro Miyaoka.

**Data curation:** Tomoko Kato-Inui, Gou Takahashi, Terumi Ono.

**Formal analysis:** Tomoko Kato-Inui, Gou Takahashi, Terumi Ono.

**Funding acquisition:** Tomoko Kato-Inui, Yuichiro Miyaoka.

**Investigation:** Gou Takahashi.

**Project administration:** Tomoko Kato-Inui, Yuichiro Miyaoka.

**Supervision:** Yuichiro Miyaoka.

**Validation:** Yuichiro Miyaoka.

**Visualization:** Tomoko Kato-Inui.

**Writing – original draft:** Tomoko Kato-Inui.

**Writing – review & editing:** Yuichiro Miyaoka.

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
