## [Decision Letter · Decision Letter 0]

26 Sep 2023

PONE-D-23-20236Fusion of histone variants to Cas9 enhances homology-directed repairPLOS ONE

Dear Dr. Miyaoka,

Thank you for submitting your manuscript to PLOS ONE. After careful consideration, we feel that it has merit but does not fully meet PLOS ONE’s publication criteria as it currently stands. Therefore, we invite you to submit a revised version of the manuscript that addresses the points raised during the review process.

I again apologize for the delay. As you can see, these reviews are outwardly disparate. After consideration of these reviews it is clear that he common positive theme is the focus on the improvement in the fusion with H2A.1.  I do agree that further work as proposed by reviewer , as well as the changes recommended by me as well as Reviewer 2 is indeed necessary for a convincing demonstration of this positive effect.   The most important issues (listed in order of priority) for both a rigorous scientific argument and for comprehension of the data are:

1. Test for locus dependency by testing multiple (2-3) sites for the H2A.1-Cas9 fusion induced HDR increase. This analysis is critical to rule out context dependency.

2. Analyze and report on off-target effects of the Cas9 fusions and comment on the interference any effects on interpretation of the data.

3. The presentation of the data should be reworked as bar graphs as indicated by Reviewers 1 and 2. I also found the data as reported difficult to interpret which impedes the evaluation of the manuscript..

4. The methods need to be described in more detail and not depend on references or previous work. During my reading of the paper, I found the digital PCR assay difficult to interpret without further explanation.You should also list all the g RNAs and reference them in a Table.

5. Check carefully for clarity in presentation. There were numerous issues including the use of acronyms (PAM) that should be spelled out in the text and unclear descriptions such as "pathogenic variants".

6. In the Introduction, the effects of the C-terminus of histones in DNA repair should also be referenced. 

We look forward to receiving your revised manuscript.

Kind regards,

Arthur J. Lustig, PhD

Academic Editor

PLOS ONE

Journal Requirements:

 "T. K-I.

19K06631

Japan Society for the Promotion of Science

https://www.jsps.go.jp

NO

T.K-I.

Takeda Science Foundation

https://www.takeda-sci.or.jp

NO

T.K-I.

Uehara Memorial Foundation

http://www.ueharazaidan.or.jp

NO

Y.M.

17H04993

Japan Society for the Promotion of Science

https://www.jsps.go.jp

NO

Y.M.

20H03442

Japan Society for the Promotion of Science

https://www.jsps.go.jp

NO"

"NO authors have competing interests"

Reviewers' comments:

Reviewer's Responses to Questions

**Comments to the Author**

1. Is the manuscript technically sound, and do the data support the conclusions?

Reviewer #1: Yes

Reviewer #2: Partly

2. Has the statistical analysis been performed appropriately and rigorously? 

Reviewer #1: Yes

Reviewer #2: No

3. Have the authors made all data underlying the findings in their manuscript fully available?

Reviewer #1: Yes

Reviewer #2: No

4. Is the manuscript presented in an intelligible fashion and written in standard English?

Reviewer #1: Yes

Reviewer #2: Yes

5. Review Comments to the Author

Reviewer #1: The authors test the effect of histone fusions with Cas9 on the ratio of DSB repair by HDR or NHEJ using two gRNAs in two target genes. They find that fusions with H2Ax and H2A1 yield in an improved HDR/NHEJ ratio mainly due to reduced NHEJ repair. The results clarify the utility of histone fusions on improving precise gene editing and will be of interest for the gene editing community aiming for precise knockin alleles.

Minor topics: it is difficult to deduce the actual ratio of HDR/NHEJ from the bars in the Figures. The ratio should be also calculated and expressed in an actual number for quantification.

It is obvious in Fig.4 that the GRN2 gRNA results in a much higher HDR activity than GRN-g2 which cuts only 2 bp apart, though on the opposite strand. The difference is very strong and an understanding of the reason could also yield clues to achieve higher HDR ratios. The authors should include this and possible reasons or future directions in the discussion.

Reviewer #2: In the manuscript titled “Fusion of histone variants to Cas9 enhances homology-directed repair”, the authors hypothesized that attaching Cas9 to phosphorylated histone H2A.X would increase HDR efficiency as it is one of initial events during repair of DNA double strand break. The authors designed various constructs with several Histone H2A variants as they did not observe increased efficiency with H2A.X. Throughout the manuscript quantitative droplet PCR was used to make all conclusions. Data presented did not validate and confirm the hypothesis. However, data presented showed H2A.1 exhibited the improved HDR/NHEJ ratio better than H2A.X. although the difference was not much but it was significant as shown by the statistical analysis. The results are interesting but premature for publication and more experiments have to be performed to validate the interesting concept proposed by the authors. Therefore, the manuscript cannot be published in PLOS one.

Specific points

1. The frequencies of HDR and NHEJ should be plotted as bar diagram to represent data more clearly, standard error and number of replicates should be mentioned. Also, the assay backgrounds should be included while representing the data. These would help in proper interpretation of the data presented.

2. The fold increase in the HDR/NHEJ ratio should be mentioned in the results section for all the experiments.

3. Did the authors check off target effects of the designed Cas9 variants? This needs to be done

4. Test, mention and discuss whether any locus specific difference was observed in the constructs which showed increase in HDR.

5. Methods have not been written in details. All details of the experiments performed in this work should be reported. Citation of earlier paper is not enough. This would improve the paper.

6. The effect of fusion of Histone H2A .1 to Cas9 on the activity of Cas 9 should be demonstrated in human cells at 2-3 different sites.

6. PLOS authors have the option to publish the peer review history of their article (what does this mean?). If published, this will include your full peer review and any attached files.

Reviewer #1: No

Reviewer #2: No

---

## [Author Response · Author response to Decision Letter 0]

12 Nov 2023

We thank Reviewers #1 and #2 for their insightful comments. While revising the manuscript based on their comments, we realized that “suppression of NHEJ” describes the positive effect of the fusion of Cas9 and histone variants more precisely. Therefore, we have changed the title of the manuscript from “Fusion of histone variants to Cas9 enhances homology-directed repair” to “Fusion of histone variants to Cas9 suppresses non-homologous end-joining”. However, the main points of our findings have not changed.

Reviewer #1: 

The authors test the effect of histone fusions with Cas9 on the ratio of DSB repair by HDR or NHEJ using two gRNAs in two target genes. They find that fusions with H2Ax and H2A1 yield in an improved HDR/NHEJ ratio mainly due to reduced NHEJ repair. The results clarify the utility of histone fusions on improving precise gene editing and will be of interest for the gene editing community aiming for precise knockin alleles.

We greatly appreciate that Reviewer #1 recognizes the value of our manuscript.

Minor topics: 

it is difficult to deduce the actual ratio of HDR/NHEJ from the bars in the Figures. The ratio should be also calculated and expressed in an actual number for quantification.

We have calculated the actual ratios of HDR/NHEJ and presented these values as superimposed bar diagrams and dot plots in each figure.

It is obvious in Fig.4 that the GRN2 gRNA results in a much higher HDR activity than GRN-g2 which cuts only 2 bp apart, though on the opposite strand. The difference is very strong and an understanding of the reason could also yield clues to achieve higher HDR ratios. The authors should include this and possible reasons or future directions in the discussion.

We appreciate Reviewer #1 for pointing this out. Thanks to the comment, we realized that we had swapped the data for RBM20-g1 gRNA and GRN-2 gRNA when we made Fig. 4B. The HDR and NHEJ activities were not much different between GRN-2 and GRN-g2 gRNAs. We apologize for this mistake and have corrected the error. 

Reviewer #2: 

In the manuscript titled “Fusion of histone variants to Cas9 enhances homology-directed repair”, the authors hypothesized that attaching Cas9 to phosphorylated histone H2A.X would increase HDR efficiency as it is one of initial events during repair of DNA double strand break. The authors designed various constructs with several Histone H2A variants as they did not observe increased efficiency with H2A.X. Throughout the manuscript quantitative droplet PCR was used to make all conclusions. Data presented did not validate and confirm the hypothesis. However, data presented showed H2A.1 exhibited the improved HDR/NHEJ ratio better than H2A.X. although the difference was not much but it was significant as shown by the statistical analysis. The results are interesting but premature for publication and more experiments have to be performed to validate the interesting concept proposed by the authors. Therefore, the manuscript cannot be published in PLOS one.

We appreciate that Reviewer #2 values the core concept of our manuscript. We have revised our manuscript to further validate the positive effects of the fusion of histone variants, thanks to very helpful comments by Reviewer #2.

Specific points

1. The frequencies of HDR and NHEJ should be plotted as bar diagram to represent data more clearly, standard error and number of replicates should be mentioned. Also, the assay backgrounds should be included while representing the data. These would help in proper interpretation of the data presented.

We have replaced the original graphs with superimposed bar diagrams and dot plots. The raw data are also in S11-15 Tables. Statistical information including the number of replicates is now more clearly stated in Materials and Methods, and Figure legends. The background noises of the assays are now included in Fig. 1C, Fig. 4C, S11 Table, and S15 Table, and we have confirmed those background noises were negligible.

2. The fold increase in the HDR/NHEJ ratio should be mentioned in the results section for all the experiments.

We have added the values of the fold decrease and increase for all the results in the manuscript. The actual values of the fold decrease and increase are also in S11-S15 Tables. 

3. Did the authors check off target effects of the designed Cas9 variants? This needs to be done.

We have identified three top predicted off-target sites for each gRNA by CRISPOR (http://crispor.tefor.net/), and have performed amplicon sequencing to monitor off-target effects in these regions. As a result, we did not observe any marked deterioration in the off-target effects by the fusion of H2A.1 to Cas9 in these potential off-target sites compared to Cas9 alone. These data are shown in S3 Fig and S16 Table. The sequences of the predicted off-target sites, the primers used for amplicon sequencing, and reagent composition and thermal cycle conditions of PCR for preparation of libraries for amplicon sequencing are listed in S7-S10 Tables. 

4. Test, mention and discuss whether any locus specific difference was observed in the constructs which showed increase in HDR.

To address whether the fusion of H2A.X or H2A.1 to Cas9 suppresses NHEJ in other targets, we have conducted additional experiments to edit the ATP7B gene and the APOE gene. The fusion of these two histone variants to Cas9 also suppresses NHEJ while keeping HDR comparable in these gene loci, although the extent of the improvements varied in different genes. In general, the HDR and NHEJ activities are highly dependent on target loci, as we observed in our previous studies (Miyaoka et al, 2016; Kato-Inui et al, 2018). The exact mechanism that causes this locus dependency is still unknown. However, the fusion of the histone variants to Cas9 can be expected to be effective in various loci.

5. Methods have not been written in details. All details of the experiments performed in this work should be reported. Citation of earlier paper is not enough. This would improve the paper.

We have added more detailed information on the genome editing target genes and mutations related to genetic disorders, the sequences of oligonucleotide donor DNAs and gRNAs, and the reagents and conditions of digital PCR in S2 Fig and S2-S6 Tables. 

6. The effect of fusion of Histone H2A .1 to Cas9 on the activity of Cas9 should be demonstrated in human cells at 2-3 different sites.

We have measured the HDR and NHEJ activities with three additional gRNAs (ATP7B-3, ATP7B-g3, and APOE-g1) in two genes. NHEJ was suppressed by fusion proteins of Cas9 and a histone variant compared to Cas9 alone for all gRNAs. We concluded that H2A.1-Cas9 is the best option based on the results from all gRNAs tested in this study.

---

## [Decision Letter · Decision Letter 1]

15 Jan 2024

Fusion of histone variants to Cas9 suppresses non-homologous end joining

PONE-D-23-20236R1

Dear Dr. Miyaoka,

We’re pleased to inform you that your manuscript has been judged scientifically suitable for publication and will be formally accepted for publication once it meets all outstanding technical requirements. Please note the following trivial change indicated by Reviwer 2. " Please lebel the Y axis in bar diagrams i Figure 1C and in the other figures whereever bar diagrams have been added."

Kind regards,

Arthur J. Lustig, PhD

Academic Editor

PLOS ONE

Additional Editor Comments (optional):

Note to authors: Please lebel the Y axis in bar diagrams inFigure 1C and in the other figures whereever bar diagrams have been added.

Reviewers' comments:

Reviewer's Responses to Questions

**Comments to the Author**

1. If the authors have adequately addressed your comments raised in a previous round of review and you feel that this manuscript is now acceptable for publication, you may indicate that here to bypass the “Comments to the Author” section, enter your conflict of interest statement in the “Confidential to Editor” section, and submit your "Accept" recommendation.

Reviewer #1: All comments have been addressed

Reviewer #2: All comments have been addressed

2. Is the manuscript technically sound, and do the data support the conclusions?

Reviewer #1: Yes

Reviewer #2: Yes

3. Has the statistical analysis been performed appropriately and rigorously? 

Reviewer #1: Yes

Reviewer #2: Yes

4. Have the authors made all data underlying the findings in their manuscript fully available?

Reviewer #1: Yes

Reviewer #2: Yes

5. Is the manuscript presented in an intelligible fashion and written in standard English?

Reviewer #1: Yes

Reviewer #2: Yes

6. Review Comments to the Author

Reviewer #1: The authors have addressed my topics for improvement in the revised version. Therefore the study of histone-Cas9 effects on NHEJ/HDR is completed.

Reviewer #2: Most of the problems in the earlier manuscript has been addressed. Please lebel the Y axis in bar diagrams i Figure 1C and in the other figures whereever bar diagrams have been added.

7. PLOS authors have the option to publish the peer review history of their article (what does this mean?). If published, this will include your full peer review and any attached files.

Reviewer #1: No

Reviewer #2: No

---

## [Editor Report · Acceptance letter]

30 Apr 2024

PONE-D-23-20236R1 

PLOS ONE

Dear Dr. Miyaoka, 

I'm pleased to inform you that your manuscript has been deemed suitable for publication in PLOS ONE. Congratulations! Your manuscript is now being handed over to our production team.

Kind regards, 

on behalf of

Dr. Arthur J. Lustig 

Academic Editor

PLOS ONE